# Delay Neural Networks (DeNN) for exploiting temporal information in event-based datasets

## Abstract

In Deep Neural Networks (DNN) and Spiking Neural Networks (SNN), the information of a neuron is computed based on the sum of the amplitudes (weights) of the electrical potentials received in input from other neurons. We propose here a new class of neural networks, namely Delay Neural Networks (DeNN), where the information of a neuron is computed based on the sum of its input synaptic delays and on the spike times of the electrical potentials received from other neurons. This way, DeNN are designed to explicitly use exact continuous temporal information of spikes in both forward and backward passes, without approximation. (Deep) DeNN are applied here to images and event-based (audio and visual) data sets. Good performances are obtained, especially for datasets where temporal information is important, with much less parameters and less energy than other models.

## 1 Introduction

Deep Neural Networks (DNN) have gained more and more in complexity, power and performance to solve highly complex tasks (Rawat and Wang, 2017). These networks abstract the functioning of biological neurons. Electrical information is integrated, computed and passed from the preceding layer to the next. As these networks can use a lot of energy (Strubell et al., 2019), and aren't biologically plausible, a new class of neural networks has emerged, Spiking Neural Networks (SNN), which tend to reproduce the spiking behavior of biological neurons.

In an SNN, each neuron is represented by an electrical membrane potential, which evolves according to incoming spikes. Once the membrane potential reaches a threshold, the neuron emits a spike and its membrane potential is usually reset. This all-or-nothing behavior reduces the number of computations because neurons possibly do not fire and thus do not activate downstream neurons. The thresholding of membrane potentials induces a discontinuity in the model, which impedes mathematical analysis and the computation of the gradient in the backpropagation algorithm and thus complicates the learning.

Time dimension can be used in different manners in SNN, depending on the model. In (Kim and Panda, 2021), using rate-coded SNN, it is possible to show that short inter-spike intervals carry information. In (Thorpe et al., 2001), in order to better account for the precise firing times of the neurons, a new coding method has emerged, namely Time-To-First-Spike (TTFS) coding. Thorpe et al. (2001) argue that the biological brain could make use of precise timing of the spikes, or of the order of arrival of each incoming spike. In TTFS coding, neurons are usually forced to spike only once, thus constraining the network to arrange spikes in time. Recently, the learning of synaptic delays has came along the will of using the time dimension (Zhang and Li, 2020; Hazan et al., 2022; Hammouamri et al., 2023; Pengfei et al., 2023). Indeed, it can be shown that, for some type of data, delays but not weights are necessary to solve temporal logic problems (Habashy et al., 2024). Moreover, temporal plasticity can be used to treat temporal information (Wang and Crus, 2024). For more theoretical analyses on the interest of synaptic delays in SNN, one can refer to (Maass, 1997; Maass and Schmitt, 1997; Thorpe et al., 2001).

We present here Delay Neural Networks (DeNN), which can be considered to be a temporal version of DNN, or an abstract SNN. As classic DNN and SNN treat electrical amplitudes (or models of), DeNN treat timing information (or a model of). In DeNN, learning happens through synaptic delays, and an important connection between any two neurons is represented by a short delay. The firing time of a neuron is computed by directly considering the impact of each presynaptic spike onto the firing time of the postsynaptic neuron instead of thresholding a membrane potential. This allows side stepping the challenge of non-differentiability faced by SNN, and using exact temporal information from input synapses in the forward and backward passes. We show results on event-based datasets for classification tasks (video and audio).

The **contributions** of this work are as follows: (i) We introduce a **new general framework for working in the temporal dimension with deep neural networks**, which can be adapted to deep neural network architectures ; (ii) In this framework, **different temporal kernels can be experimented** to learn **synaptic delays** instead of synaptic weights, with exact evaluation of the gradient in the temporal dimension ; (iii) **On benchmark datasets (images, videos and audio), DeNN obtains the same, or better performances than other models, but with much less parameters** and less energy cost, with respect to other models.

## 2 Related works

### 2.1 Temporal coding in SNNs

Learning temporal codes has a long history in computational neuroscience. We can cite, among many other approaches, Tempotron (Gütig and Sompolinsky, 2006) and ReSuMe (Ponulak and Kasiński, 2009). Tempotron presents an interesting weight synaptic learning rule for a single neuron to learn for detecting locally synchronous spikes, known as Spike-Timing-Dependent Plasticity (STDP). ReSUME focuses on making (reservoir networks of) neurons learn to reproduce template signals (instructions) encoded in precisely timed sequences of spikes. Instead, the purpose of DeNN is to follow a modernized version of Time-To-First-Spike (TTFS) temporal code implementing delays, and adapting the idea to event-based datasets. DeNN neurons learn synaptic and firing delays for firing faster or slower according to the classification error of a temporal signal in a deep learning network architecture. While DeNN is currently more oriented to global deep learning mechanisms, computational neuroscience mechanisms will help in the future to improve and to better understand the global and local learning mechanisms of DeNNs.

Mostafa (2017) was one of the first to introduce temporal coding in modern deep spiking neural networks. To do so, they derived an analytic formula to directly compute the firing times of non-leaky Integrate and Fire (IF) neurons which produce a single spike, *i.e.*, with infinite refractory period. Zhou et al. (2021) extended this algorithm to more challenging benchmark datasets in computer vision. Their work was extended to the Spike Response Model (SRM) neurons by (Comsa et al., 2020), where an equation is solved in the complex field to find solutions for the good spike timing of the neurons. With the same technique, Göltz et al. (2021) extended the work in (Mostafa, 2017) to several cases of Leaky Integrate and Fire neurons (LIF). All these models have in common a methodology to solve an equation to find the timing of the first spike from a neuron (whether it is an IF neuron, a LIF, an SRM). With the same methodology, Park et al. (2020) presented equations somewhat simpler than in (Göltz et al., 2021; Comsa et al., 2020).

Another solution is to adapt the backpropagation algorithms to event-based data (Zhu et al., 2022). Recent works (Wunderlich and Pehle, 2021; Lee et al., 2023) show that an exact computation of the gradient for these models is possible, and that rate coding and temporal coding can be related through loss functions (Zhu et al., 2023). Most of these approaches are either restricted to specific models (Mostafa, 2017; Comsa et al., 2020; Göltz et al., 2021) or to approximate gradients (Zhu et al., 2022). Among approximate gradient techniques, the surrogate gradient allows SNNs to exhibit promising results (Esser et al., 2016; Bellec et al., 2018; Neftci et al., 2019; Tavanaei et al., 2019). However, this technique remains an

approximation of the thresholding function used in the forward pass by continuous functions in the backward pass.

Another possibility that has been developed is to map classic neurons to spiking neurons, as in (Rueckauer and Liu, 2018; Park and Yoon, 2021; Kheradpisheh and Masquelier, 2020). These works allow simpler backpropagation algorithms, since they can leverage better the properties of analog networks. They are however still restricted to one model of neuron and can hardly be generalized to other models.

### 2.2 Synaptic delays

To achieve temporal coding, Single spike SNNs (Mostafa, 2017; Zhang et al., 2019; Zhou et al., 2021; Comsa et al., 2020; Göltz et al., 2021; Park et al., 2020) intuitively seem also to be a good framework to learn delays between neurons. Modelling the delays into neural networks can be tracked back at least to 1989 with Time-Delay Neural Networks (TDNN) (Waibel et al., 1989). In this work, connections have several synaptic terminals, each with its own fixed delay and variable weight thus leading to an exploding number of parameters. Bohte et al. (2002) derived an approach called Spike Prop, with the same architecture for synaptic terminals and the same drawbacks in terms of memory and computations.

Simpler delay-based models have then been developed. In (Schrauwen and Van Campenhout, 2004; Wu et al., 2006; Shrestha and Orchard, 2018; Hammouamri et al., 2023; Pengfei et al., 2023) one trainable delay was implemented for each synaptic connection, alongside with synaptic weights. In (Taherkhani et al., 2015a;b), single output neurons are trained to fire a spike train at desired times. Also, Taherkhani et al. (2015b) only allow delays to be increased, which seems biologically implausible. Zhang and Li (2020) presented an interesting joint synaptic delay-weight plasticity algorithm, and confronted it to a real-world dataset for speech recognition. More recent works tend to get rid to the single spike constraint, as it is not reliable for event based datasets (Yu et al., 2023; Hammouamri et al., 2023; Pengfei et al., 2023; Grappolini and Subramoney, 2023; Deckers et al., 2024; Wang, 2024).

In most of these delay-based works, every synapse has two parameters: a weight and a delay, effectively doubling the memory cost with respect to analog networks. To the best of our knowledge, (Hazan et al., 2022) is the only work presenting a weightless spiking neural network, where learning happens only through synaptic delays. The authors used a learning rule derived from Spike-Timing-Dependent Plasticity (STDP), and confronted their network to a classic image classification task. However, their network exhibits low accuracy and memory performances.

## 3 Methods

### 3.1 Forward pass

In our network, instead of weights $w_{ij}$, we use positive delays which represent a simple time delay between two presynaptic and postsynaptic neurons. In DNN, if a connection between two neurons is important (inversely not important), its weight $|w_{ij}|$ is high (inversely low), while in DeNN this corresponds to a small (inversely high) delay $|d_{ij}|$. In order to ensure that the delays $d_{ij}$ stay positive during the back-propagation steps, we effectively declare signed-delays $d_{ij}^s \in \mathbb{R}$, and then compute the delay as a simple Gaussian of the signed-delays:

$$d_{ij} = \exp\left(-\left(\frac{d_{ij}^s}{\sigma_j}\right)^2\right) \qquad (1)$$

where $\sigma_j$ is a parameter learnt for each neuron.

**DeNN's neurons generic definition simply consists of Equation 2**. The firing time of postsynaptic neuron $j$ is computed as a function of the spiking times of the preceding layer ($t_i > 0$) and synaptic delays:

$$t_j = \sum_i f(t_i, d_{ij}^s) \qquad (2)$$

The function $f$ represents the synaptic impact of presynaptic neuron $i$ onto the spiking time of postsynaptic neuron $j$, $t_j$. Essentially, for any synaptic input received, which is excitatory (resp. inhibitory), every neuron $j$ fires earlier (resp. later). In DeNN, low (inversely high) spiking times would correspond to high (inversely low) activations in DNN.

**Many DeNN's neuron functions $f$ can be derived from Equation 2**. The following has been chosen after considering arrival times $t_i + d_{ij}$ for presynaptic neuron $i$ and postsynaptic neuron $j$:

$$t_j = \sum_i sign(d_{ij}^s)\Big[\kappa(t_i + d_{ij}) - \kappa(t_i + 1)\Big] \tag{3}$$

where $\kappa$ is a strictly decreasing positive function representing the impact of the incoming spike onto the firing time of the postsynaptic neuron. The term $-\kappa(t_i + 1)$ represents an incompressible delay because it decreases the activity of the synapse, and tackles a discontinuity at $d_{ij} = 1$ (see Section A for more details). The sign of the signed-delay, between a presynaptic neuron $i$ and a postsynaptic neuron $j$, is taken into account to represent the type of the synapse: whether it is an excitatory synapse (negative synapses, to decrease the time of activation), or an inhibitory one (positive synapse). We found that a satisfactory kernel $\kappa$ was the exponential kernel $\kappa(x) = e^{-x}$. Thus, with this kernel, each spike is exponentially more important than subsequent ones. This corresponds to a balanced mix between TTFS and rank-order coding (Thorpe et al., 2001). In that sense, DeNN is a temporal abstraction of the behavior of biological neurons. Note that any continuous positive decreasing function can be used for kernel $\kappa$, but the exponential one happens to work well in practice.

### 3.2 Standardization and temporal ReLU function

Neurons' spike times are standardized, with average and standard deviation:

$$t_j = \sum_i sign(d_{ij}^s)\Big[\kappa(t_i + d_{ij} - t_q) - \kappa(t_i + 1 - t_q)\Big] \tag{4}$$

where $t_q$ is the activation time of the $q$-th quantile after which every incoming spike is canceled.

It is possible then to cancel (or not) every neuron that fires after some value (typically the median of the spiking times in the layer), forcing them to an infinite time. This process is equivalent to a simple lateral inhibition, where the first neurons of a layer to spike in time impede the neighbouring slower ones. It is equivalent to sending a signal to the neurons of the layer after the spike corresponding to the $q$-th quantile to make them silent. With $q = 1$, all spikes are fired, this corresponds to a slow regime (Figure 1, left). With $q = 0.5$, all spikes after the median (or the average in case of gaussian distribution, see Section G.1 for an experimental illustration) are silent, and this corresponds to the fast regime (Figure 1, right). **In software systems, the DeNN equations simply become:**

$$t_j = \sum_i sign(d_{ij}^s)\Big[\kappa(z_i + d_{ij}) - \kappa(z_i + 1)\Big]$$
$$z_j = std(t_j) \tag{5}$$

where $std$ is the standardization process, where we subtract the mean and divide by the standard deviation of the distribution. As shown in Section B, taking $\kappa(z_i + d_{ij})$ or $\kappa(t_i + d_{ij} - t_q)$ is "almost" equivalent, up to a division by standard deviation. For the fast regime ($q = 0.5$), a temporal ReLU can be defined as:

$$TempReLU(z) = \begin{cases} z & \text{if } z < 0 \\ +\infty & \text{otherwise} \end{cases} \tag{6}$$

### 3.3 Events Preprocessing: event2time algorithm

When event-based datasets consist of images obtained from event-based cameras, pixel-level intensity changes are captured as events. Since the number of events increases with the temporal resolution of the camera, the number of events can get large.

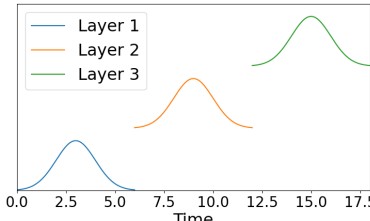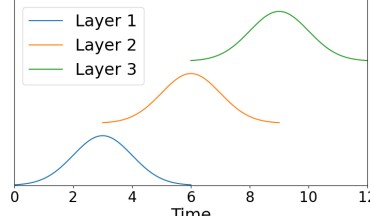

Figure 1: Slow (left) and fast (right) regimes of the DeNN. Each layer outputs spikes after an integration phase, which duration is calibrated by $q$. If $q = 1$, then each layer has to wait until every neuron of the preceding layer has emitted a spike, which corresponds to the slow regime. To reduce the latency of the model, it is possible to decrease $q$ (fast regime), so that each layer can ignore the slowest neurons of the preceding layer.

To deal efficiently with event-based datasets, an algorithm for the pre-processing of events, inspired from (Zeigler, 2004), was developed (Dagang, 2022). This algorithm greatly reduces the number of events in the datasets while tracking the most relevant pixels' activity. In event-based datasets, a data consists of three coordinates $(t, p, \mathbf{x})$ where $t$ is the time of the event, $p \in \{-1, +1\}$ is the polarity of the event, and $\mathbf{x}$ is the position of the pixel in the image. Our algorithm, called *event2time*, accumulates events on each cell over time. Each cell $i$ stores the timing of the events in a list $L_i$, and when more than $2rN$ ($0 < r < 1$, for $N$ total pixels in the image, and two polarities) cells are active (*i.e.*, have stored one or more events), an array is built with:

$$t_i = \frac{\max L_i - \min L_i}{\#L_i}; z_i = std(t_i) \tag{7}$$

where $\#L_i$ is the size of the list. We reiterate this process on subsequent events until we reach the end of the sample. The above equation transforms strongly active cells into fast cells, and conversely poorly active cells into slow cells. This process is represented in Figure 2. A sample $S$ is thus represented by $M$ images $I_1, ..., I_M$ presented successively to the network, which emits a prediction. Each neuron emits at most one spike per image $I_s$. The prediction of the network is stored for each image $I_s$, and, at image $I_M$, the pseudo-probability that the sample is of class $c$ is computed given the past information with a temporal softmin:

$$P(S = c | I_1, I_2, ..., I_M) = \pi_c = \frac{\sum_{s=1}^{M} e^{-z_c[I_s]}}{\sum_{s=1}^{M} \sum_{j=1}^{K} e^{-z_j[I_s]}} \tag{8}$$

where $z_c[I_s]$ represents the standardized activation time for the $c$-th output neuron at input image $I_s = I_1, ..., I_M$. For audio-based datasets, we first applied the *speech2spike* (Stewart et al., 2023) algorithm to transform the raw audio into events, and then applied our *event2time* algorithm.

## 3.4 BACKWARD PASS

In order to train our network, we use the traditional backpropagation algorithm, where gradient descent is performed on the signed-delays $d_{ij}^s$. Our aim is to decrease important delays, instead of increasing important weights. The classic learning rule of backpropagation is used:

$$d_{ij}^s \leftarrow d_{ij}^s - \eta \frac{\partial L}{\partial d_{ij}^s} \tag{9}$$

where $0 < \eta < 1$ is the learning rate. Note that, in practice, we used the Adam optimizer, whose gradient formula is slightly different but can be found in (Diederik and Ba, 2014). The loss function $L$ at the end of the sequence is the traditional cross-entropy loss for classification tasks. And we have:

$$\frac{\partial L}{\partial d_{ij}^s} = \frac{\partial L}{\partial \pi_j} \frac{\partial \pi_j}{\partial z_j} \sum_{s=1}^{M} \frac{\partial z_j[I_s]}{\partial t_j[I_s]} \frac{\partial t_j[I_s]}{\partial d_{ij}} \frac{\partial d_{ij}}{\partial d_{ij}^s} \tag{10}$$

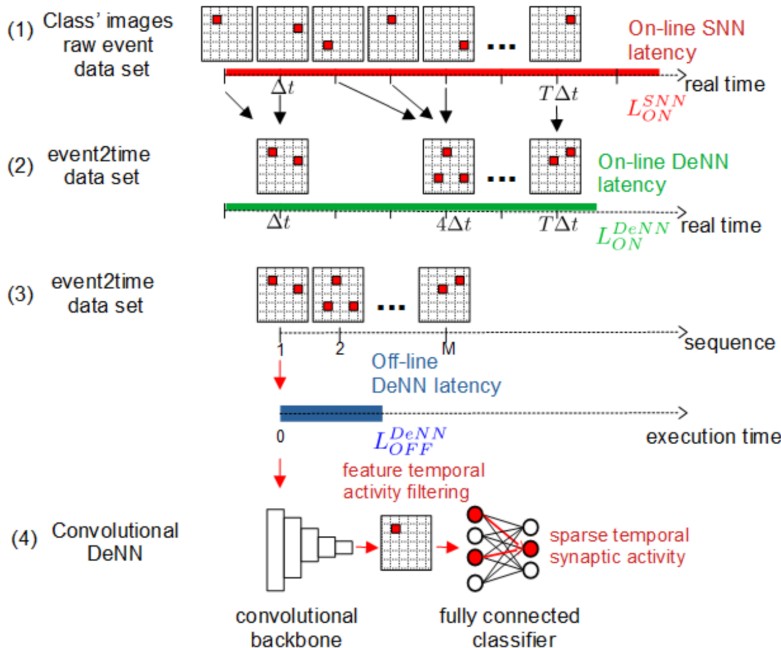

Figure 2: DeNN full pipeline: (1) Events arrive every $\Delta t$ timestep, where $\Delta t$ is the precision of the neuromorphic camera, and are used such as by SNN. (2) Using event2time algorithm, events are aggregated over $T\Delta t$ timesteps, for one sample, and fed to the network in an on-line manner. (3) After all events are computed, the sequence of $M$ images can be simulated much faster jumping from one image to the other without waiting for every $\Delta t$ timestep, giving rise to an off-line latency. (4) Feed-forward network used for each image.

where every term is well defined and is derived from differentiable functions, which allows us to directly use generic libraries for automatic differentiation, such as PyTorch (Paszke et al., 2017). More details on the computation of the gradient can be found in Appendix C.

### 3.5 Short and Long Term Memory

Our preprocessing algorithm captures the temporal dynamics of each sample over short windows of time. Consider that, on average, each input cell presents an activation (or an event) every $\Delta m$ timesteps. If the dataset has $N$ input cells, and we want $2rN$, active cells on each image $I_s$, then on average, each image $I_s$ represents a short-term window of $\Delta m2rN$ timesteps, and the network acquires short term memory. However, for some data, there is a need for longer term memory. Longer term memory is assigned to each neuron of the network as follows:

$$\delta^h = z_j[s] - z_j[s-h], \ h = s - \nu, ..., s$$

$$z_j[s] \leftarrow z_j[s] + \sum_{h=s-\nu}^{s} \alpha_j^h sign(\delta^h)[\exp(-|\delta^h|) - 1] \tag{11}$$

where $\alpha_j^h \in [-1, 1]$ is a learnt parameter for neuron $j$. Hence, each cell has a long-term memory equals to $\nu\Delta m2rN$, with $\nu$ a constant hyperparameter. More details can be found in Section E.

## 4 Results

### 4.1 Performance on benchmark datasets

To show that our network is able to tackle temporal data, we confronted it to the event-based version of MNIST dataset (N-MNIST (Orchard et al., 2015)), to the DVS Gesture

dataset (Amir et al., 2017) which represents hand movements, and to the Google Speech Command (GSC) dataset (Warden, 2018), which is a speech recognition dataset. In order to allow for larger comparisons with other models, we also confronted our model to the MNIST (Le Cun et al., 1998) and CIFAR-10 (Krizhevsky, 2009) datasets. Every training details and parameter values can be found in Section D. Results and comparison to the state of the art models are shown in Table 1. For each dataset, we compare to models that either have the best performance or the best accuracy to parameter ratio (see Table F.1 for more comparisons on performance). While performances are preserved for visual tasks, DeNN improves the performances in audio tasks, where temporality is important. These performances are achieved with architectures much lighter than for other models. Furthermore, when possible, we computed the average number of computations per sample in different models, by multiplying the reported firing rate to the number of synapses (see Appendix F for more details). We show that it is possible to achieve good performances with fewer computations for MNIST, CIFAR-10, N-MNIST and DVS-Gesture.

## 4.2 Energy Consumption

Based on the theoretical computational complexity of Table F.1, energy consumption results have been computed in Table F.2 for the neuromorphic supercomputer of the Human Brain Project, SpiNNaker (Painkras et al., 2013). We show that DeNN consumes less energy than other best performing models on all datasets, except maybe for DVS gesture, where we could not find the firing rate of the best performing model. Furthermore, as discussed in Section F, these good results will be improved in the near future. Exponential function operations are essential for machine learning. This is leading to increased research in electrical engineering to reduce hardware energy consumption. Impressive power reduction has been achieved recently on the electronic devices used by neuromorphic computers (Costa et al., 2023).

## 4.3 Choice of kernel $\kappa$

The class of model presented in this work is general enough to work with any kernel, as long as it is a decreasing positive function. Although the kernel that works best is the exponential $\kappa(x) = \exp(-x)$, we found satisfying result with the inverse $\kappa(x) = x^{-1}$. To avoid double negatives (when both $x$ and $d_{ij}^s$ are negative), we shifted the Gaussian curve (after standardization) by three units to the right and clipped the (standardized) spike times to a minimum of 0.001. We reported an accuracy of 96.62% for the MNIST dataset, and 97.76% for the neuromorphic version. Note that in order to compare only the change in kernel, we used the same architectures and hyperparameters as for $\kappa(x) = \exp(-x)$. It should be possible to obtain even better results by adapting the architectures and parameters of the model. Also, for resources reason, these simulations were performed only on small datasets.

## 4.4 Time for event-based models and event-based datasets

To show how a DeNN uses the temporal information in event-based datasets, Figure 3 depicts the probabilities that a sound $S$ is of class $c$ given the past, for a sample of sound drawn from the GSC dataset. The probabilities clearly evolve with time and inputs, as the networks get more information about the stimulus. Figure 4b shows the evolution of accuracy on GSC dataset for different values of long-term memory. With hyperparameter memory length value $\nu = 5$ (Equation 11), the model already classifies correctly about 93% of inputs, and it crosses 96% at $\nu \geq 10$.

We also note that the activity of neurons accelerates in the direction of the movement for the DVS-Gesture dataset, as illustrated in Figure 5. We computed the difference $\delta$ (see Equation 11) between neurons of the first layer's feature maps created after the input of a sample of the DVS-Gesture dataset. We found that, in the direction of a movement (here a right-hand counter clockwise), neurons tend to decrease their relative timing spike to others (i.e. $z_j[s] < z_j[s-1]$), while they increase afterwards. This is consistent with the hypothesis that time in visual system encodes speed and direction of stimuli (Saleh Vahdatpour and Zhang, 2024). To further explore the time dimension in the DVS-Gesture dataset, an experiment was run where the total number of timesteps during inference was truncated. The

Table 1: Comparison of performance accuracy on benchmark datasets. When possible, we computed the exact number of active synapses in the model. A synapse is considered active if it transmits a value other than 0. See Appendix F for more details on the average number of computations based on computational complexity. W stands for weight training, and D for delay.

| Model | # Parameters / active | Avg FLOPS per object | Top-1% Accuracy | Acc / log(#Params) ratio |
|---|---|---|---|---|
| **MNIST** | | | | |
| (Zhang et al., 2020), W | 635,200 | - | **98.40%** | 7.8472 |
| (Kheradpisheh and Masquelier, 2020), W | 317,600 / 35,755 | 35,755 | 97.40% | 9.2899 |
| DeNN ($q = 1$), D | 79,400 / 14,804 | 14,804 | 97.46% | 10.1493 |
| DeNN ($q = 0.5$), D | 79,400 / **8,135** | **8,135** | 97.43% | **10.8208** |
| **CIFAR-10** | | | | |
| (Zhou et al., 2021), W | $54.2 \cdot 10^6$ | $249.4 \cdot 10^6$ | **92.68%** | 5.2043 |
| (Park and Yoon, 2021), W | $33.6 \cdot 10^6$ | - | 91.90% | 5.3029 |
| DeNN ($q = 1$), D | $5.8 \cdot 10^6$ / $\mathbf{2.3 \cdot 10^6}$ | $61.5 \cdot 10^6$ | 90.59% | **6.1843** |
| DeNN ($q = 0.5$), D | $5.8 \cdot 10^6$ / $\mathbf{1.4 \cdot 10^6}$ | $\mathbf{43.2 \cdot 10^6}$ | 87.09% | 6.1539 |
| **N-MNIST** | | | | |
| (Fang et al., 2021), W | - | - | **99.61%** | - |
| (Zhu et al., 2022), W | 35,800 | - | 99.39% | 9.5006 |
| DeNN ($q = 0.5$), D | 15,696 / **11,788** | **665,262** | 98.06% | **10.5666** |
| **DVS-Gesture** | | | | |
| (Cordone et al., 2021), W | 13,992 | $< 10 \cdot 10^6$ | 92.01% | 9.6383 |
| (Man et al., 2023), W | - | - | **98.23%** | - |
| DeNN ($q = 0.5$), D | 19,216 / **7,895** | $\mathbf{5.8 \cdot 10^6}$ | 97.57% | **10.8725** |
| **GSC** | | | | |
| (Bittar and Garner, 2024), W+D | $1.5 \cdot 10^6$ | - | 97.05% | 6.6667 |
| (Deckers et al., 2024), W+D | 610,000 | $\sim \mathbf{3.45 \cdot 10^6}$ | 95.69% | 7.1833 |
| DeNN ($q = 1$), D | **175,467** | $\sim 3.6 \cdot 10^6$ | **97.73%** | 8.0934 |

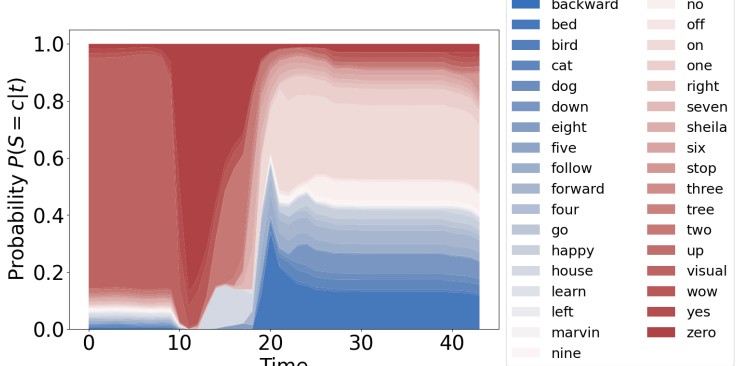

Figure 3: Graph of probabilities $p$ for each class of the dataset given the past at each timestep $t$, for a sound of the GSC dataset ("Off").

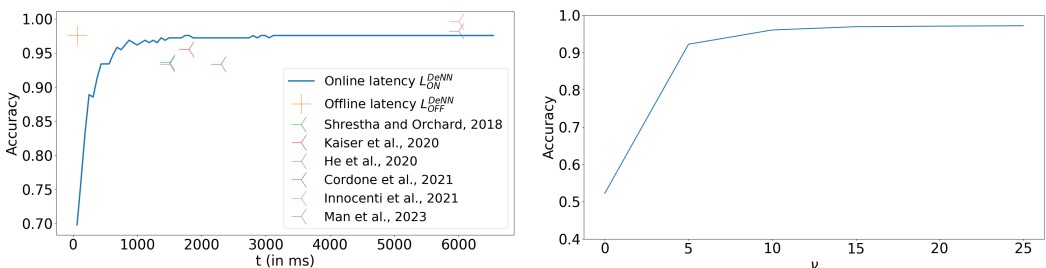

(a) Accuracy obtained with different maximum timesteps for a model trained on full samples, for the DVS-Gesture dataset.

(b) Accuracy obtained on the GSC dataset with models with short ($\nu = 0$) to long-term memory.

Figure 4: Ablation studies for DVS-Gesture and GSC

results are presented in Figure 4a. It is shown that a DeNN can achieve good performance in a few timesteps, which hints that the time dimension in the DVS-Gesture dataset might not be the most important dimension. Indeed, samples in this dataset are composed of periodic movements. Hence, only a few periods are required to really discriminate between movements.

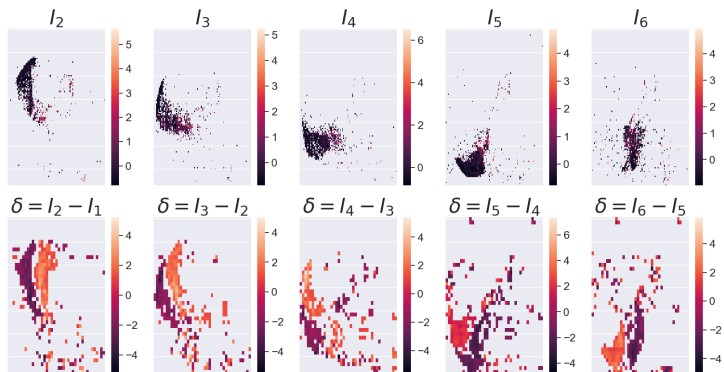

Figure 5: Top line: Input images $I_2$ to $I_6$ from a right-hand counter clockwise movement of the DVS-Gesture dataset, obtained after application of our preprocessing algorithm. Bottom line: Differences $\delta$ between neurons of the first convolutional layer's feature maps at images $I_s$ and $I_{s-1}$. Darker pixels indicate faster neurons between two timesteps. Note that neurons in the range $[-1, 1]$ are canceled for clarity of image, leaving us with neurons where the difference is significant enough.

### 4.5 LATENCY

Figure 2 shows that while SNN requires a discrete time simulation in real time, with a $\Delta t$ step (see step (1)), DeNN can use a discrete event simulation without any $\Delta t$ step (see step (3)). As shown in step (2), DeNN on-line latency is theoretically lower than SNN on-line latency, because a DeNN does not have to wait for the last $\Delta t$ step to be consumed after last image at $T\Delta t$ step. For the online latency, a timestep on the Gesture dataset represents, on average, 0.6227 seconds of the sample, and 0.6099 seconds for the N-MNIST dataset. On-line property relies on the fact that input events are received in real time (e.g., by an event-based camera of a robot/car). However, as shown in step (3), if on-line learning in real time is not required, DeNN allows as fast as possible discrete event simulation. All the images are fed to the network in a sequence leading to smaller off-line latency $L_{OFF}^{DeNN} = 0.066$ seconds on a NVIDIA GeForce RTX 3080 Laptop GPU; for the Gesture dataset. This is particularly interesting for example to train cars for automated driving by off line simulation. This allows greatly reducing training time and energy consumption. We show on Figure 4a the comparison with other SNN models for the Gesture dataset. We note that models reporting better accuracy are also models with higher latencies. Note that this as-fast-as-possible discrete event simulations do not depend on slow or fast regime. Although the slow regime requires receiving all the spikes from the previous layer to compute the output spike times of the layer, it does not require waiting for any real time discrete time step $\Delta t$. Even in a slow regime, layers can be simulated as fast as possible. The only difference of the fast regime is that it reduces the number of computations and execution time, while slightly decreasing performances.

### 4.6 FUNCTIONING OF THE NETWORK

Figure 6 shows the activity of synapses connecting the neurons in the intermediate layer of the classifier to the eighth output neuron, for the MNIST dataset, for images of the digit 8. In a DeNN (left), very few synapses are of extreme importance to make the output neuron spike much before the others, while others synapses are silent. In an ANN (right), there

must be a balance between inhibition and excitation. The main difference between the ANN and the DeNN stems in the fact that, in our network, paths that are irrelevant do not show any activity, while in an ANN they are inhibited. This shows how DeNN is able to drastically decrease the number of computations with respect to other models. All couples (neuron, digit) have been computed and are available on figures I.1 (for MNIST) and I.2 (for CIFAR-10).

This behaviour might be due to the peculiar derivative of $t_j$ with respect to the parameter $d_{ij}^s$. As shown in Figure C.1, the derivative is close or equal to zero for almost every input $z_i$. Even when the input is sufficiently strong (reminding that negative $z$ is quicker input than others), the derivative plunges toward zero at $d_{ij}^s = 0$, which means that synapses will have trouble changing signs. Some will get asymptotically close to zero (negative or positive), while never being able to change sign.

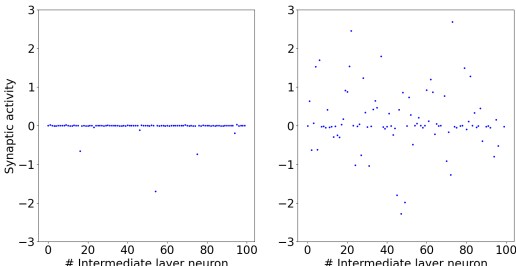

Figure 6: Synaptic activity is defined as $sign(d_{ij}^s)[\kappa(z_i + d_{ij}) - \kappa(z_i + 1)]$ for a DeNN(on the left) and $w_{ij}x_i$ for an ANN (on the right). Each dot is a synapse.

## 5 CONCLUSION

A new class of neural networks was presented. These networks are able to treat temporal information, arguing they are, alongside (Hazan et al., 2022), the first network to be fully temporally coded, by taking into account the spike order's importance, as presented in (Thorpe et al., 2001; Liu et al., 2023). Indeed, the model in (Hazan et al., 2022) is temporally coded, but lacks performances. Finally, DeNN are able to achieve satisfying results on tasks with much less parameters than other models, and with less computations. On audio task, where temporality is important, DeNN demonstrates better results than state-of-the-art models. We also show that DeNN use less energy than other models. On datasets where temporal dimension is crucial, DeNN use less energy with better performance results than the adaptive LIF model used in (Bittar and Garner, 2022; 2024; Deckers et al., 2024).

Since the equations for the DeNN are general, it is possible to adapt any (continuous) architecture to the DeNN with few efforts. We were indeed able to very easily apply our model to convolutional architectures, and try out different kernels $\kappa$ fairly easily. Conversion of existing architectures to spiking one would be an interesting development for DeNN, as in (Man et al., 2023; 2024). A limitation can be discussed for long-term memory hard window definition. The memory cost increases with $\nu$ for sequences longer than a few seconds. However we have to note that thanks to our preprocessing algorithm, one timestep in our model is equal to $\Delta m 2 r N$ timesteps in real time. With $\nu$, the memory goes to $\nu \Delta m 2 r N$, as explained in Section 3.5. It should be possible to encode sequences longer than a few seconds with few adaptations to the preprocessing algorithm (for example, with a bigger $r$). Also, we show in Table F.2 that even with $\nu = 25$, the DeNN uses less energy than the adLIF used in (Deckers et al., 2024), which uses coupled dynamic equations, one for the short memory (membrane potential), and one for the long-term memory (recovery variable).

This work should open new perspectives to the field of temporal coded networks, as the need to better take into account the temporality of information, especially with the rising use of event-based cameras and brain-inspired learning.

## Data and code availability

The datasets used in this study are publicly available on the Internet. Code is available from the corresponding author on request.

## Acknowledgments

The authors would like to thank Andrew Rowley, researcher from the Human Brain Project in the School of Computer Science at the University of Manchester, and in charge of SpiNNaker developments, for his profound insights of the energy consumption of the SpiNNaker chip.

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

## SUPPLEMENTARY

## A   SPIKING TIME CONTINUITY

We show on Figure A.1 how the term $-\kappa(z_i + 1)$ corrects a discontinuity at $d_{ij} = 1$. If $d_{ij} = 1$, $d_{ij}^s = 0$.

$$
\begin{aligned}
sgn(d_{ij}^s)kappa(t + d_{ij}) \xrightarrow[d_{ij}^s \to 0^-]{} -kappa(t + 1) \\
sgn(d_{ij}^s)kappa(t + d_{ij}) \xrightarrow[d_{ij}^s \to 0^+]{} kappa(t + 1)
\end{aligned}
\tag{12}
$$

Since $\kappa$ is a strictly decreasing positive function, $-kappa(t + 1) < kappa(t + 1)$, hence there is a discontinuity at $d_{ij} = 1$.

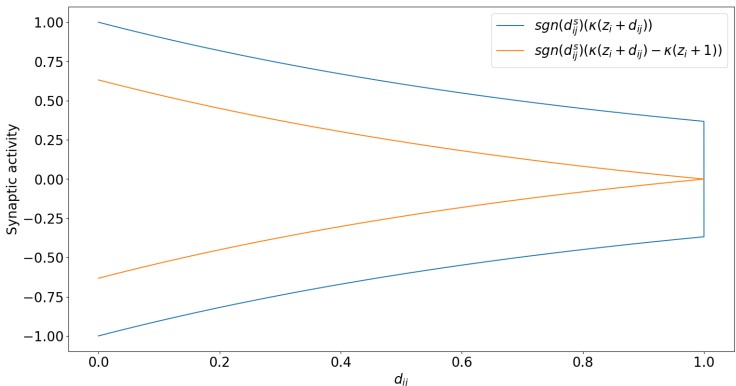

Figure A.1: Synaptic activity with and without the correction term, as a function of delays $d_{ij}$.

## B   SPIKING TIME STANDARDIZATION

Taking $\kappa(z_i + d_{ij})$ in Equation 4, or $\kappa(t_i + d_{ij} - t_q)$, in Equation 5, is "almost" equivalent, up to a division by standard deviation.

Taking the median quantile $t_q = t_{0.5}$, if $t_i$ are normally distributed (which they are, see Figure G.1), we have:

$$
\begin{aligned}
X := z_i + d_{ij} = \frac{t_i - t_{0.5}}{\sigma} + d_{ij} \sim N(d_{ij}, 1) \\
Y := t_i - t_{0.5} + d_{ij} \sim N(d_{ij}, \sigma).
\end{aligned}
\tag{13}
$$

The difference we get in evaluating $\kappa(t_i + dij - t_{0.5})$ and $\kappa(\frac{t_i - t_{0.5}}{\sigma} + d_{ij})$ depends on $\sigma$. It can get high the further $\sigma$ gets from 1. However, dividing by $\sigma$ allows for better numerical stability in the evaluation of $\kappa$ kernel, because it allows every layer to operate in the same range of values. Moreover, the division will reflect during backpropagation algorithm, because it will appear in the derivatives. Hence, the two mechanisms are "almost" equivalent.

## C   BACKWARD COMPUTATION

The loss function at the end of the sequence is the traditional cross-entropy loss for classification tasks:

$$
L = -\sum_c^K target_c \log(\pi_c)
\tag{14}
$$

with

$$P(S = c|I_1, I_2, ..., I_T) = \pi_c = \frac{\sum_s^T e^{-z_c[I_s]}}{\sum_s^T \sum_j^K e^{-z_j[I_s]}} \tag{15}$$

and

$$\frac{\partial L}{\partial \pi_c} = -\frac{1}{\pi_c} \tag{16}$$

$$\frac{\partial \pi_c}{z_l} = \begin{cases} \pi_c(\pi_c - 1) & \text{if } l = c \\ \pi_c \pi_l & \text{if } l \neq c \end{cases} \tag{17}$$

For differentiating the $z_j$ variable, we need to take into account the fact that $z_j$ depends on the input $I_s$. Thus:

$$\frac{\partial z_j}{\partial d_{ij}^s} = \sum_s^T \frac{\partial z_j[I_s]}{\partial t_j[I_s]} \frac{\partial t_j[I_s]}{\partial d_{ij}} \frac{\partial d_{ij}}{\partial d_{ij}^s} \tag{18}$$

with $\kappa(x) = e^{-x}$ as kernel, we have:

$$\frac{\partial t_j}{\partial d_{ij}^s}[I_s] = \frac{2|d_{ij}^s|}{\sigma^2} d_{ij} e^{-(z_i[I_s] + d_{ij})} \tag{19}$$

represented on Figure C.1. In order to tackle vanishing and exploding gradients which could arise, a gradient normalization is implemented at each layer, using the Frobenius norm of the gradient matrix.

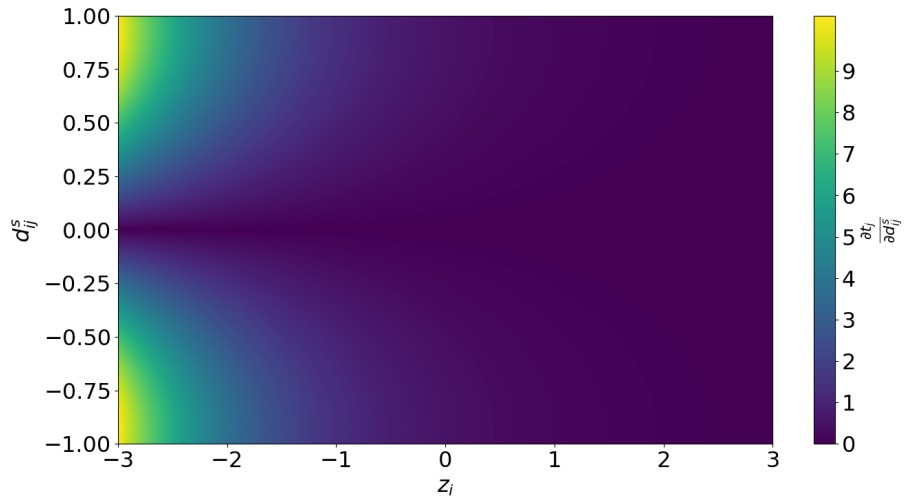

Figure C.1: Derivative of $t_j$ with respect to parameter $d_{ij}^s$.

## D  TRAINING DETAILS

Widespread PyTorch library (Python version 3.8.6, PyTorch 1.10.0, (Paszke et al., 2017)) has been used for achieving fair performance comparisons and also for its high capabilities. Experiments were conducted on a RTX 2080 Ti GPU.

Models' architectures are described in Table D.1 and parameters used for learning on each dataset are detailed in Table D.2 in Appendix. The Adam algorithm (Diederik and Ba, 2014) with default parameters is used to perform the backpropagation.

Table D.1: Architectures used for the different datasets. A convolutional layer described as 8c5s2 means 8 filters of size 5x5 with a stride of 2. A minpool layer with 2x2 kernels and a stride of 2 is described as p2s2.

| Dataset | Model Architecture |
|---|---|
| MNIST | 784-100-10 |
| CIFAR-10 | VGG-9 |
| N-MNIST | 8c5s2 - 16c3s1 - p2s2 - 32c3s1 - 32c3s1 - p2s2 - 10 |
| DVS-Gesture | 8c7s3 - 16c5s2 - p2s2 - 32c3s1 - 32c3s1 - p2s2 - 11 |
| GSC | 60-256-256-256-35 |

Table D.2: Parameter values for each dataset.

| Parameters | MNIST | CIFAR-10 | N-MNIST | DVS-Gesture | GSC |
|---|---|---|---|---|---|
| Batch size | 4096 | 512 | 16 | 16 | 700 |
| Learning Rate | 0.001 | 0.001 | 0.001 | 0.001 | 0.001 |
| LR Scheduler | - | - | - | - | CosineAnnealing |
| $\Delta$ | - | - | 4 | 4 | 1 |
| $r$ | - | - | 0.05 | 0.05 | 0.1 |
| Seed | 22756400 | 76446569 | 94240977 | 98074194 | 36887311 |
| s2s channels | - | - | - | - | 30 |
| s2s threshold | - | - | - | - | 0.75 |
| $\nu$ | - | - | 0 | 0 | 25 |

## E  LONG-TERM MEMORY

Long-term memory is added to the network thanks to this equation:

$$\delta^h = z_j[s] - z_j[s-h], \; h = s - \nu, ..., s$$
$$z_j[s] \leftarrow z_j[s] + \sum_{h=s-\nu}^{s} \alpha_j^h sign(\delta^h)[\exp(-|\delta^h|) - 1] \tag{20}$$

The mechanism is represented on Figure E.1. The main mechanism is that, if neuron $j$ is faster than its neighbors ($\delta^h > 0$ or $z_j[s-h] < z_j[s]$), then it gets a small boost ($z_j[s]$ is decreased). Conversely, if it was slower than its neighbors, then it gets a small punishment ($z_j[s]$ is increased). Note that $\alpha_j^h$ is a learnable parameter in the range $[-1, 1]$, so the network can decide to reverse boost and punishment.

## F  COMPUTATIONAL COMPLEXITIES & ENERGY COST

It is described here how to obtain the average number of computations per object derived for Table 1.

For sparse convolutions used in (Cordone et al., 2021), the average number of active sites $n_a$ (Graham et al., 2018) per layer per timestep can be computed with the ratio of the number of spikes in a layer to the number of timesteps. However, we are unable to know the spatial distribution of activated sites. For a lower bound estimation, it can be assumed that each of them was visited exactly once, thus underestimating the number of computations. The lower bound for average number of computations per layer and timestep is then $n_a C_o^\ell C_o^{\ell-1}$, with $C^\ell$ the number of channels of the convolutional layer, and the total number of computations per one sample of the dataset is obtained by summing over all layers and all timesteps, and is approximately of $1.2 \cdot 10^6$ computations. An upper bound can be obtained by assuming that active sites are densely grouped on a square of side $\sqrt{n_a}$, which gives, when summing over all layers and timesteps, approximately $10 \cdot 10^6$ computations.

SpiNNaker power consumptions can be found in (Painkras et al., 2013). The energy consumption per instruction can be inferred as follows. The power of a SpiNNaker's chip is

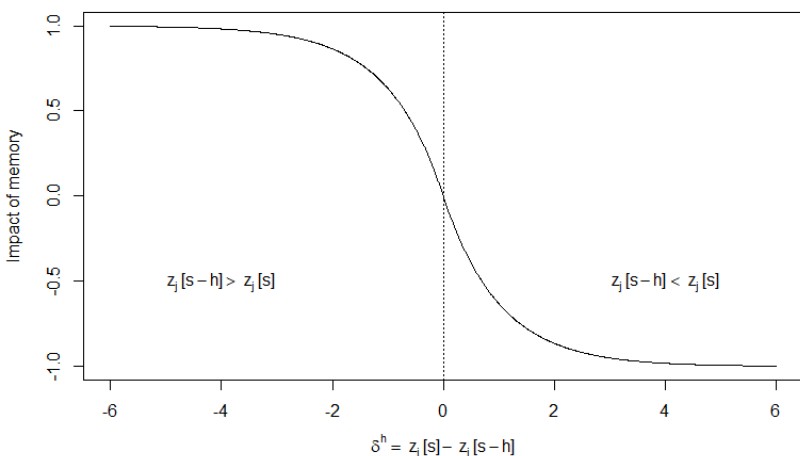

Figure E.1: Representation of the term $sign(\delta^h)[\exp(-|\delta^h|) - 1]$ in Equation 20

1W peak. The idle total chip power is then made up of the idle chip power (0.36W) plus the SDRAM (0.170W), i.e., 0.53W. Of the remaining active power (1W - 0.53W = 0.47W), each link could use 0.063W and there are six, so 0.378W. Leaving, 0.47W - 0.378W = 0.092W, for core activity. There are 18 cores, so this is 0.0051W per core. Each clock cycle takes $5 \cdot 10^{-9}$ seconds, so the energy consumption per clock cycle is then $2.56 \cdot 10^{-11} J$.

From cores' technical documentation ARM (2004), it can be found that each multiplication requires 2 clock cycles and each addition/subtraction 1 clock cycle. Using SpiNNaker own code (Partzsch et al., 2017), based on fixed-point calculations (Partzsch et al., 2017), each exponential function computation requires 95 clock cycles[1].

Table F.2 presents the energy consumption on SpiNNaker neuromorphic supercomputer.

Note that neuromorphic chips functioning and electronic architecture (Orchard et al., 2021; Akopyan et al., 2015; Mayr et al., 2019) are based on discrete event simulations. Many discrete time (synchronous) models, presented in Table F.2, have the energy consumption advantage of not using exponential functions. However, their spike time is not exact, being up to the time step precision. In discrete event (asynchronous) models (like DeNN), spike times are exact but their computations come with the price of exponential function computations. However, many machine learning approaches require exponential functions. Currently, it is a problem taken seriously by electronics engineers. Energy consumption cost of an exponential function computation has recently been reduced to the insignificant cost of $3.63 \cdot 10^{-12} J$ for VLSI CMOS technology (Costa et al., 2023), which is used by neuromorphic computers.

## G    Distribution of neurons' spike times before standardization

We show on figure G.1 that the distributions of the spike times inside a layer, before the standardization process, is gaussian.

## H    Functioning of the network

We show on Figure H.1 how a convolutional filter follows the activity in time.

---

[1]Energy consumption results are extracted from a very interesting discussion with Andrew Rowley, researcher from the Human Brain Project in the School of Computer Science at the University of Manchester, and in charge of SpiNNaker developments.

Table F.1: We set $n^\ell$ the number of neurons in layer $\ell$, $n_s^\ell$ the number of spiking neurons in layer $\ell$, while $T$ is the total number of timesteps. $I$ and $C$ represent sets of causal spikes (Mostafa, 2017; Göltz et al., 2021; Comsa et al., 2020). For rate coding networks, $\tau$ represents the ratio of the total number of spikes observed in $T$ timesteps to the total number of neurons (Datta et al., 2021). For convolutional layers, $H_o^\ell, W_o^\ell, C_o^\ell$ and $k^\ell$ are the height and width of the feature maps, the number of channels and the size of the kernel.

| Dataset Neural coding | Model | Computational complexity | Accuracy |
|---|---|---|---|
| **MNIST** | | | |
| TTFS | (Zhang et al., 2020) | $\mathcal{O}\big(n^l[T-1+T(n_s^{\ell-1}+1)]\big)$ | 98.40% |
| TTFS | (Kheradpisheh et al., 2022) | $\mathcal{O}\big(n^\ell[T-1+T(n_s^{\ell-1}+1)+n_{\bar{s}}^{\ell-1}]\big)$ | 97.00% |
| TTFS | (Comsa et al., 2020) | $\mathcal{O}\big(n^{\ell-1}+|I|n^\ell\big)$ | 97.96% |
| TTFS | (Im et al., 2022) | $\mathcal{O}\big(n^\ell[T-1+T(n_s^{\ell-1}+1)]\big)$ | 96.00% |
| TTFS | (Oh et al., 2022) | $\mathcal{O}\big(T[n^\ell(n_s^{\ell-1}+1)]\big)$ | 96.90% |
| TTFS | (Mostafa, 2017) | $\mathcal{O}\big(n^{\ell-1}+2|C|n^\ell\big)$ | 97.55% |
| TTFS | (Göltz et al., 2021) | $\mathcal{O}\big(n^{\ell-1}+|C|n^\ell\big)$ | 97.10% |
| TTFS | (Kheradpisheh and Masquelier, 2020) | $\mathcal{O}\big(n^\ell[T(n_s^{\ell-1}+1)+T-1+n_{\bar{s}}^{\ell-1}]\big)$ | 97.40% |
| Temporal | DeNN ($q=1$) | $\mathcal{O}\big(n^\ell[n^{\ell-1}+2]\big)$ | 97.46% |
| Temporal | DeNN ($q=0.5$) | $\mathcal{O}\big(n^\ell[n_s^{\ell-1}+2]\big)$ | 97.43% |
| **CIFAR-10** | | | |
| TTFS | (Datta et al., 2021) | $\mathcal{O}\big(C_o^\ell H_o^\ell W_o^\ell[\tau C_o^{\ell-1}C_o^\ell k^{\ell^2}+2T-1]\big)$ | 91.41% |
| TTFS | (Zhou et al., 2021) | $\mathcal{O}\big(C_o^{\ell-1}k^{\ell^2}[H_o^\ell W_o^\ell k^{\ell^2}(|C|+1)+C_o^\ell H_o^\ell W_o^\ell]\big)$ | 92.68% |
| TTFS | (Park et al., 2020) | $\mathcal{O}\big(H_o^\ell W_o^\ell C_o^\ell[\tau C_o^{\ell-1}k^{\ell^2}+1]\big)$ | 91.43% |
| Temporal | DeNN ($q=1$) | $\mathcal{O}\big(C_o^\ell H_o^\ell W_o^\ell[C_o^{\ell-1}k^{\ell^2}+2]\big)$ | 90.59% |
| Temporal | DeNN ($q=0.5$) | $\mathcal{O}\big(C_o^\ell H_o^\ell W_o^\ell[\tau C_o^{\ell-1}k^{\ell^2}+2]\big)$ | 87.09% |
| **N-MNIST** | | | |
| Rate | (Lee et al., 2016) | $\mathcal{O}\big(n^\ell[\tau n^{\ell-1}+T(2+n^\ell)]\big)$ | 98.74% |
| Rate | (Wu et al., 2018) | $\mathcal{O}\big(n^\ell[\tau n^{\ell-1}+T]\big)$ | 98.78% |
| Rate | (Shrestha and Orchard, 2018) | $\mathcal{O}\big(H_o^\ell W_o^\ell C_o^\ell[T+\tau C_o^{\ell-1}k^{\ell^2}]\big)$ | 99.20% |
| Rate | (Kaiser et al., 2020) | $\mathcal{O}\big(H_o^\ell W_o^\ell C_o^\ell[T+\tau C_o^{\ell-1}k^{\ell^2}]\big)$ | 96% |
| Rate | (Jin et al., 2018) | $\mathcal{O}\big(n^\ell[\tau n^{\ell-1}+T]\big)$ | 98.84% |
| Rate | (Lee et al., 2020) | $\mathcal{O}\big(H_o^\ell W_o^\ell C_o^\ell[T+\tau C_o^{\ell-1}k^{\ell^2}]\big)$ | 99.09% |
| Rate | (Cheng et al., 2020) | $\mathcal{O}\big(H_o^\ell W_o^\ell C_o^\ell[\tau(C_o^{\ell-1}k^{\ell^2}+k_\omega^2)+T)]\big)$ | 99.45% |
| Rate | (Fang et al., 2020) | $\mathcal{O}\big(H_o^\ell W_o^\ell C_o^\ell[T+\tau C_o^{\ell-1}k^{\ell^2}]\big)$ | 99.39% |
| Rate | (He et al., 2020) | $\mathcal{O}\big(n^\ell[\tau n^{\ell-1}+T]\big)$ | 98.28% |
| Rate | (Fang et al., 2021) | $\mathcal{O}\big(H_o^\ell W_o^\ell C_o^\ell[T+\tau C_o^{\ell-1}k^{\ell^2}]\big)$ | 99.61% |
| Temporal | DeNN ($q=0.5$) | $\mathcal{O}\big(C_o^\ell H_o^\ell W_o^\ell[\tau C_o^{\ell-1}k^{\ell^2}+2]\big)$ | 98.06% |
| **DVS Gesture** | | | |
| Rate | (Shrestha and Orchard, 2018) | $\mathcal{O}\big(H_o^\ell W_o^\ell C_o^\ell[T+\tau C_o^{\ell-1}k^{\ell^2}]\big)$ | 93.64% |
| Rate | (Kaiser et al., 2020) | $\mathcal{O}\big(H_o^\ell W_o^\ell C_o^\ell[T+\tau C_o^{\ell-1}k^{\ell^2}]\big)$ | 95.54% |
| Rate | (Fang et al., 2020) | $\mathcal{O}\big(H_o^\ell W_o^\ell C_o^\ell[T+\tau C_o^{\ell-1}k^{\ell^2}]\big)$ | 96.09% |
| Rate | (He et al., 2020) | $\mathcal{O}\big(n^\ell[\tau n^{\ell-1}+T]\big)$ | 93.40% |
| Rate | (Fang et al., 2021) | $\mathcal{O}\big(H_o^\ell W_o^\ell C_o^\ell[T+\tau C_o^{\ell-1}k^{\ell^2}]\big)$ | 97.57% |
| Rate | (Cordone et al., 2021) | $\mathcal{O}\big(Tn_a^t C_o^\ell C_o^{\ell-1}\big)$ | 92.01% |
| Temporal | DeNN ($q=0.5$) | $\mathcal{O}\big(C_o^\ell H_o^\ell W_o^\ell[\tau C_o^{\ell-1}k^{\ell^2}+2]\big)$ | 97.57% |
| **GSC** | | | |
| Temporal | (Hammouamri et al., 2023) | - | 95.35% |
| Temporal | (Bittar and Garner, 2024) | - | 97.05% |
| Temporal | (Deckers et al., 2024) | - | 95.69% |
| Rate | (Wang et al., 2024) | - | 92.90% |
| Rate | (He et al., 2024) | - | 87.33% |
| Rate | (Boeshertz et al., 2024) | - | 93.33% |
| Temporal | DeNN ($q=1$) | $\mathcal{O}\big(n^\ell[n^{\ell-1}+2+\nu]\big)$ | 97.73% |

# I  SYNAPTIC IMPACTS OF SPIKES

Figure I.1 presents the synaptic impact of each spike of the intermediary layer onto the output layer, for the fully connected DeNN for solving the MNIST task, and in Figure I.2 the same matrix is shown for the CIFAR-10 dataset.

Table F.2: Energy consumption of models. All symbols are the same as in Table F.1.

| Model | Energy Consumption | On SpiNNaker |
|---|---|---|
| **MNIST** | | |
| (Zhang et al., 2020) | $T\tau n^\ell n^{\ell-1}(2ADD + 2MUL + IF)$ | $114T\tau\mu J$ |
| (Kheradpisheh and Masquelier, 2020) | $T\tau n^\ell n^{\ell-1}(ADD + 3IF)$ | $532\mu J$ |
| DeNN ($q=1$) | $n^\ell(\tau n^{\ell-1}(2EXP + 3ADD) + 2MUL + 5ADD) + 2MUL$ | $73\mu J$ |
| DeNN ($q=0.5$) | $n^\ell(\tau n^{\ell-1}(2EXP + 3ADD) + 2MUL + 5ADD) + 2MUL$ | **$40\mu J$** |
| **CIFAR-10** | | |
| (Zhou et al., 2021) | $C_o^\ell H_o^\ell W_o^\ell(3MUL + ADD[2\tau C_o^{\ell-1}k^{\ell^2} + 1] + EXP[\tau k^{\ell^2}C_o^{\ell-1}] + IF)$ | $620,190\mu J$ |
| DeNN ($q=1$) | $C_o^\ell H_o^\ell W_o^\ell(\tau C_o^{\ell-1}k^{\ell^2}(2EXP + 3ADD) + 2MUL + 5ADD) + 2MUL$ | $381,371\mu J$ |
| DeNN ($q=0.5$) | $C_o^\ell H_o^\ell W_o^\ell(\tau C_o^{\ell-1}k^{\ell^2}(2EXP + 3ADD) + 2MUL + 5ADD) + 2MUL$ | **$232,161\mu J$** |
| **N-MNIST** | | |
| (Zhu et al., 2022) | $TC_o^\ell H_o^\ell W_o^\ell(ADD(\tau c_o^{\ell-1}k^{\ell^2} + 3) + 2MUL + 2EXP + IF)$ | $4301(0.9\tau + 1)\mu J$ |
| (Fang et al., 2021) | $TC_o^\ell H_o^\ell W_o^\ell(ADD(\tau C_o^{\ell-1}k^{\ell^2} + 3) + 2MUL + IF)$ | $13,687\tau + 375\mu J$ |
| DeNN ($q=0.5$) | $T[C_o^\ell H_o^\ell W_o^\ell(\tau C_o^{\ell-1}k^{\ell^2}(2EXP + 3ADD) + 2MUL + 5ADD) + 2MUL]$ | **$11,616\mu J$** |
| **DVS Gesture** | | |
| (Fang et al., 2021) | $TC_o^\ell H_o^\ell W_o^\ell(ADD(\tau C_o^{\ell-1}k^{\ell^2} + 3) + 2MUL + IF)$ | $404,604\tau + 11,267\mu J$ |
| DeNN ($q=0.5$) | $T[C_o^\ell H_o^\ell W_o^\ell(\tau C_o^{\ell-1}k^{\ell^2}(2EXP + 3ADD) + 2MUL + 5ADD) + 2MUL]$ | $312,476\mu J$ |
| **GSC** | | |
| (Bittar and Garner, 2024) | $Tn^\ell(ADD[\tau(\frac{n^\ell}{2} + n^{\ell-1}) + 3] + 4MUL + COMP + 2EXP + 2MUL + 2ADD)$ | $39,4812(\tau + 0.21)\mu J$ |
| (Deckers et al., 2024) | $Tn^\ell(ADD[\tau 2n^{\ell-1} + 8] + 7MUL + COMP)$ | $25,005\mu J$ |
| DeNN ($q=1$) | $T[n^\ell(\tau n^{\ell-1}(2EXP + 3ADD) + MUL(\nu + 2) + \nu EXP + 5ADD) + 2MUL]$ | **$20,715\mu J$** |

Table F.3: Comparison of datasets, models and architectures in time and space dimensions.

| | | | Model | | | |
|---|---|---|---|---|---|---|
| | | | ANN | IF | LIF | adLIF - DeNN |
| | Time Dim / Spatial Dim | | Null | Poor | Medium | Strong |
| Architect. | Fully Connected | Null | | | | |
| | | Poor | | | | GSC |
| | | Medium | MNIST | | | |
| | Convolutional | Strong | CIFAR | | Gesture | |

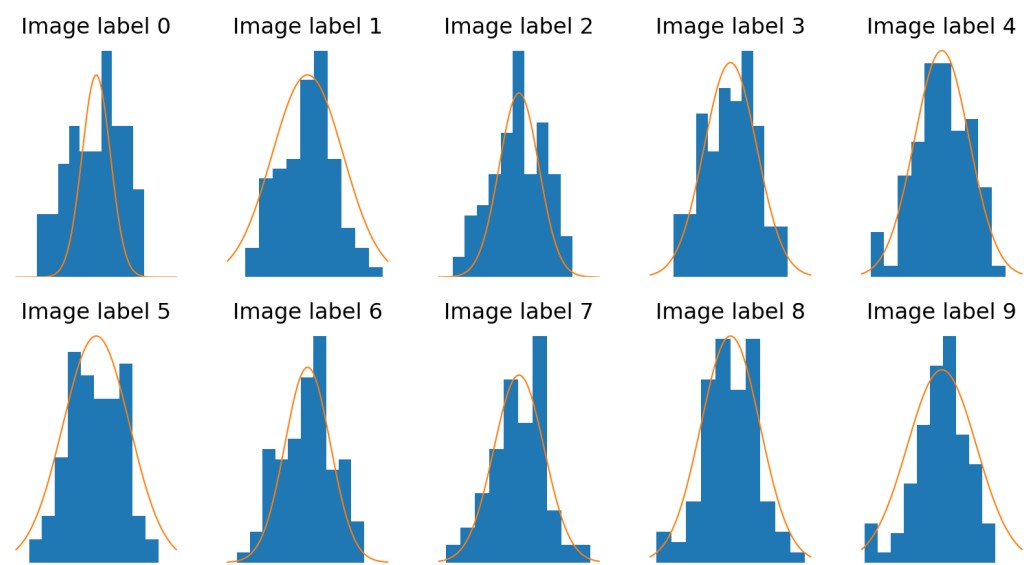

Figure G.1: Distribution of neurons' spike times before standardization for the hidden layer of a network trained on the MNIST dataset, averaged over each class of images. Theoretical probability density function plot in solid orange line.

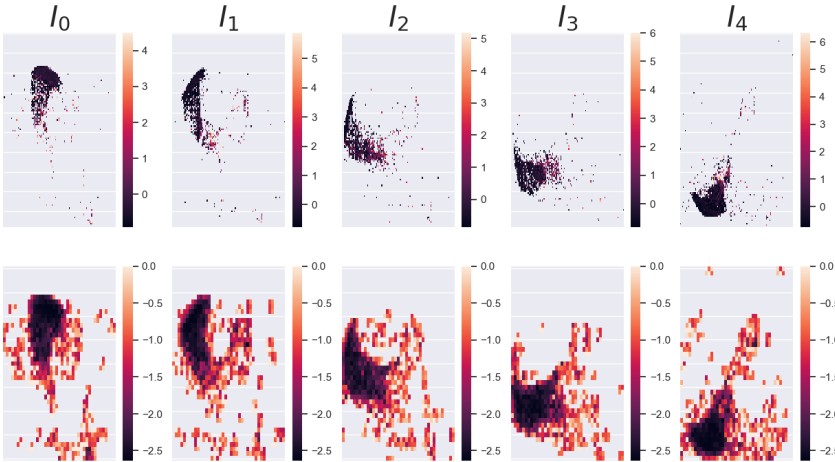

Figure H.1: Top line: input images $I_s$. On bottom line: feature maps showing that the filter follows in time the areas that have the most activity.

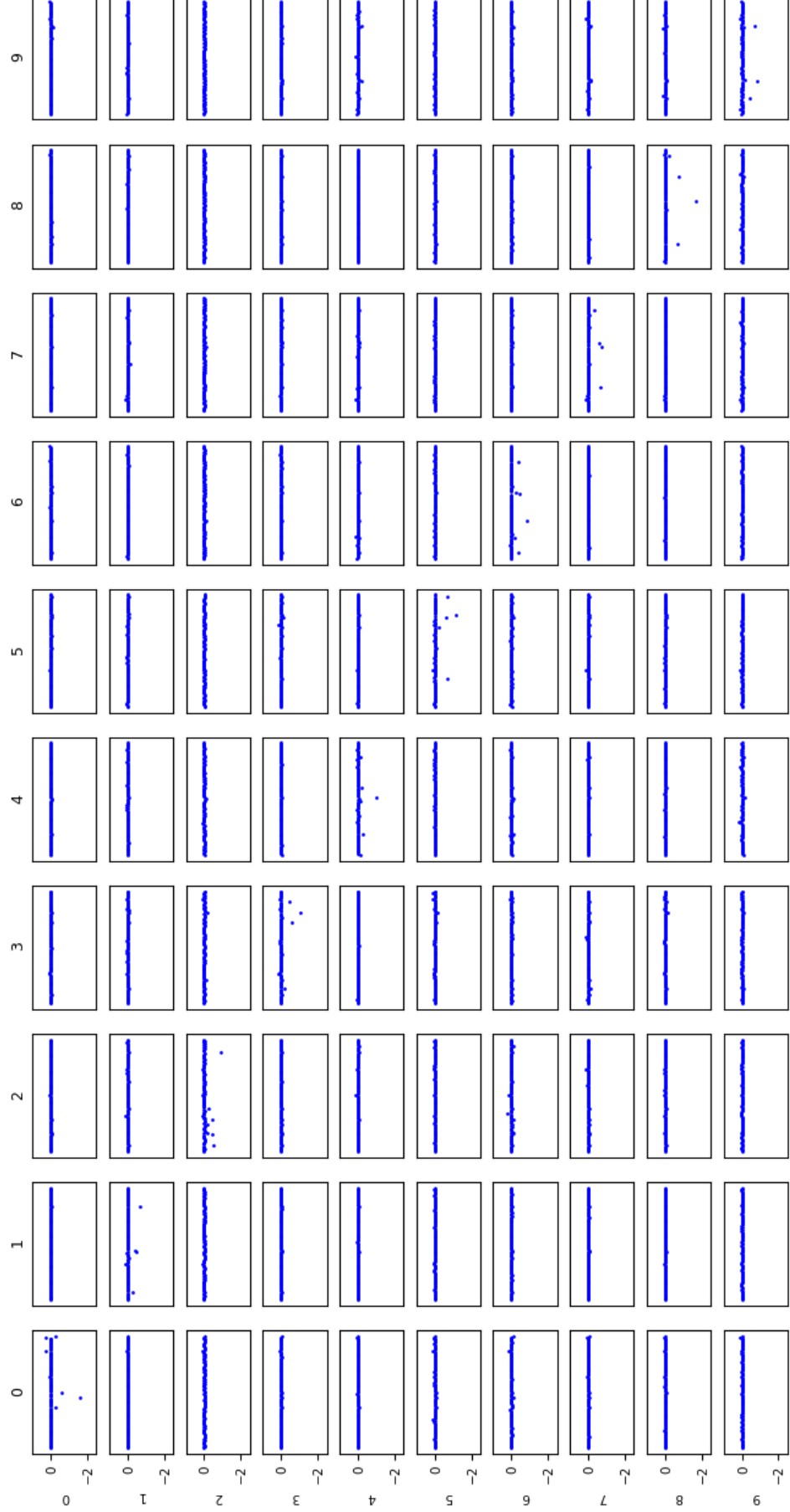

Figure I.1: This grid of 10x10 graphs represents the 10 output neurons in columns for each image category in line, for the MNIST dataset. Each point on each graph represents the average synaptic impact of one intermediate neuron onto the output neuron, the image considered.

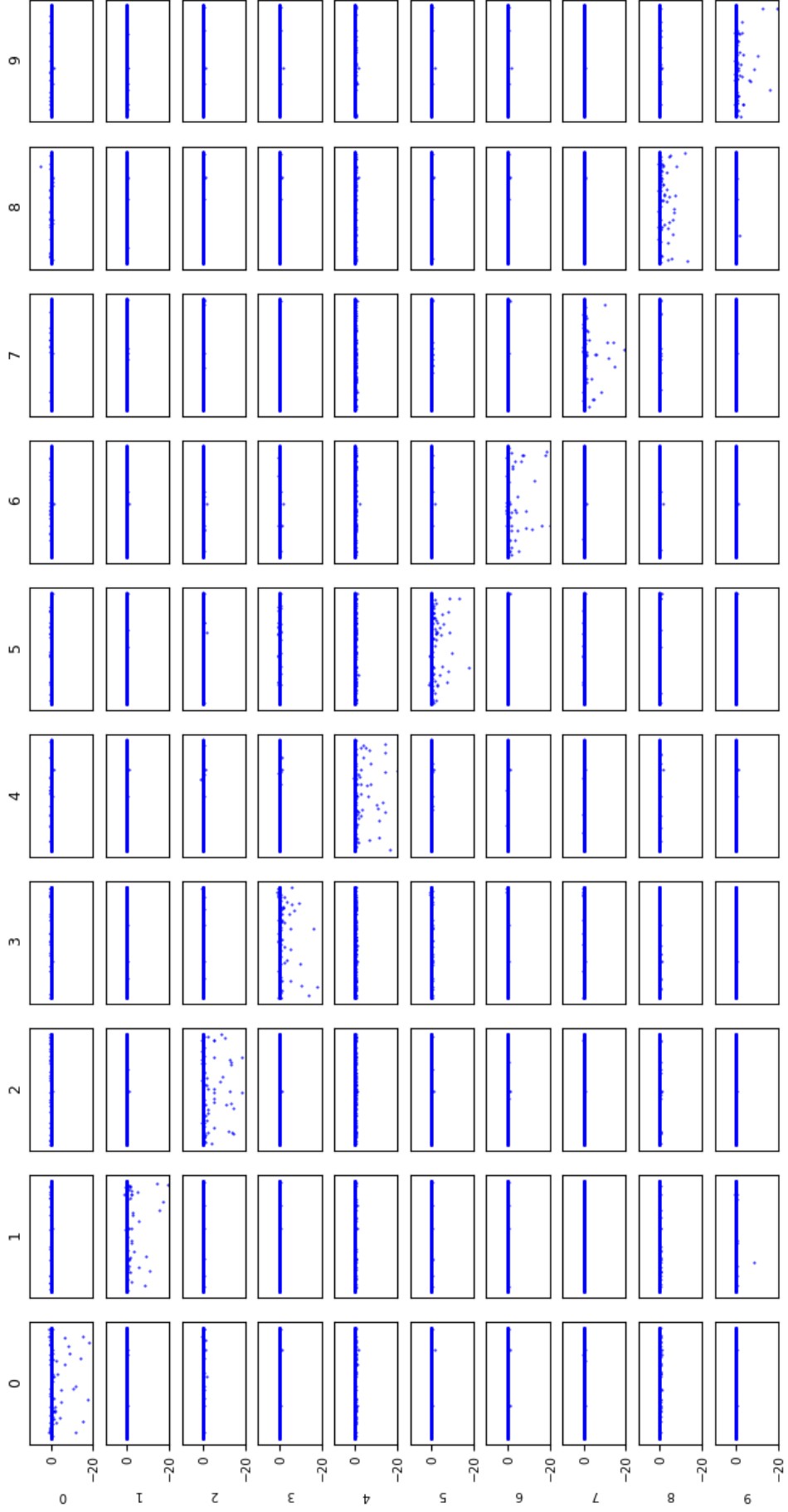

Figure I.2: This grid of 10x10 graphs represents the 10 output neurons in columns for each image category in line, for the CIFAR10 dataset. Each point on each graph represents the average synaptic impact of one intermediate neuron onto the output neuron, the image considered.

