# OpenReview forum: "Delay Neural Networks (DeNN) for exploiting temporal information in event-based datasets"
_ICLR.cc/2025/Conference — Submitted to ICLR 2025_

### Official Review · Reviewer_GDtt · 2024-11-01

**Soundness:** 3
**Presentation:** 2
**Contribution:** 2
**Rating:** 5
**Confidence:** 4

**Summary:**

This work proposes a new class of neural networks (DeNN, Delay Neural Networks), which employs delay mechanism to exactly compute temporal information of event-based data in both forward and backward pass. Besides, an event-to-time data preprocessing algorithm is developed. It achieves competitive performance in some computer vision and audio tasks, with smaller parameter count and less number of computations compared to other baselines.

**Strengths:**

1. This work opens up a successful practice in purely delay-based spiking neural networks without traditional weights, achieving competitive performance on some existing vision and audio benchmarks.
2. This work realizes compact model size while maintaining model performance.
3. By ignoring the firing of a proportion of neurons, this work provides a promising and general paradigm of sparse learning with accurate gradient computation.

**Weaknesses:**

1.	Presentations in Part 1 Introduction and Part 2 Related Works are a bit of loose and limited.

(1) Analysis of the reasons for abandoning traditional weights, as well as the effectiveness of delay mechanism against weight and membrane potential mechanism, is not sufficiently insightful.
In fact, comparison of the two mechanism’s expressive capacity is well worth thorough discussion.
Please provide a theoretical analysis at least. I will also be glad to see some empirical study of this topic.

(2) When discussing related works, different manners of spike encoding(i.e. rate coding, latency coding and delta modulation, etc) in weight and membrane potential mechanism settings, their representative works(e.g. [2307.01694]Spike-driven Transformer, or other baselines), as well as there pros and cons compared to DeNN are not fully covered.

2.	In Table 1, DeNN underperforms the SOTA models on CV benchmarks, while outperforms the SOTA models on the audio benchmark GSC. It is likely that DeNN may have some strengths to excel in audio tasks, or it may have some weaknesses in CV settings. The behaviour of DeNN when faced different modalities of data, and the reasons behind this phenomenon is not fully discussed.
Please provide a theoretical analysis of this topic.

**Questions:**

1. Will DeNN's advantage of parameter count, number of computations, and proportions of active parameters degradate when scaling?
2. From my perspective of view, the formula of postsynaptic neuron's spiking time (i.e. formula (5)) is similar to MAC(Multiply Accumulate) in weight and membrane potential mechanism, and the function of time in DeNN is also similar to that of analog activation value in artificial neural network. Can you tell me their difference in essence?
3. The energy consumption of different kinds of computation may not be the same, and energy efficiency is an important topic in brain-inspired computing. Have you measured the average theoretical energy consumption of DeNN against other SNN models?
4. In Table 1, on N-MNIST dataset, the proportion of active parameters is approximately 75%, and DeNN is performing dense rather than sparse computation. Will DeNN be forced to activate most of its neuron, to achieve competitive performance in some circumstances?

---

### Official Review · Reviewer_BPcv · 2024-11-03

**Soundness:** 2
**Presentation:** 1
**Contribution:** 2
**Rating:** 3
**Confidence:** 4

**Summary:**

This paper trains SNNs with purely synaptic delay, while mainstream methods train SNNs with synaptic weights.

**Strengths:**

Training an SNN with synaptic delays instead of weights is an interesting try. Few people have tried something like this.

**Weaknesses:**

1. The font seems to be wrong.
2. There are many symbols in this work, I recommend the authors provide a table illustrating the meaning of each symbol.
3. In the experiments, Acc / #Params is not a fair indicator. Achieving a linear increase in accuracy may require an exponential increase in the number of parameters. The authors should provide results under the same network architecture.
4. Figure 3 might be redrawed, since there are too many colors which are too close to each other. By the way, is this figure plotting one sample in the dataset?
5. Table 1 is too wide and there are some grammar mistakes like "where #*Li* is the of size of the list." in line 236.

**Questions:**

1. I am not sure for the event2time algorithm. Does it split events into frames by time or number of events and accumulate them? Or it converts the events to a single spike according to the event intensity?
2. Since the paper uses $t_j$ to represent the firing time of neuron $j$, does the algorithm limit the number of output spikes to 1 for each neuron?
3. In Eq.(7), does $s$ mean one time step? In the domain of SNNs, $t$ is often used in this case instead. By the way, $\sum_s^T$ should be replaced by $\sum\_{s=0}^T$ or $\sum_{s=1}^T$ (depending on which time step the simulation starts) in Eq.(7).
4. In Eq.(4), I am not sure for $t_q$: Will $t_i+1-t_q$ be negative? In most cases, spike response kernels $\kappa(t)$ are meaningful only when $t>=0$.

---

### Official Review · Reviewer_9G8f · 2024-11-03

**Soundness:** 2
**Presentation:** 2
**Contribution:** 2
**Rating:** 5
**Confidence:** 4

**Summary:**

DeNNs are proposed as a temporal variant of DNNs or an abstract form of SNNs. Unlike traditional models that focus on electrical amplitudes, DeNNs emphasize timing information. Learning in DeNNs occurs through synaptic delays, allowing for the direct computation of firing times based on presynaptic spikes, thus avoiding the non-differentiability issue faced by SNNs. The paper presents a new framework for integrating temporal dimensions into deep learning architectures. This framework allows for experimentation with different temporal kernels to learn synaptic delays, achieving competitive performance on benchmark datasets (images, videos, and audio) with fewer parameters than existing models.
This introduction sets the stage for understanding how DeNNs can effectively leverage temporal information in event-based datasets, addressing limitations of both DNNs and SNNs.

**Strengths:**

1.Efficient Use of Temporal Information: DeNN utilizes the exact continuous temporal information of spikes during both forward and backward passes, allowing for a more accurate representation of temporal dynamics compared to traditional models that rely on amplitude-based information.

**Weaknesses:**

1.High Computational Cost for Exponential Functions: One of the significant limitations of DeNN is the computational cost associated with calculating exponential functions, particularly on hardware devices and neuromorphic systems. This can restrict the model's applicability in resource-constrained environments, where efficient computation is crucial. This paper computes the active parameters and the total parameters, what is the advantage of active parameters? could the remaining parameters be pruned after training?

2.Memory Costs for Long Sequences: While DeNN can effectively utilize long-term memory to enhance performance, this comes at the cost of increased memory requirements. For sequences longer than a few seconds, the memory cost can become prohibitively high, which may limit its use in applications involving extended temporal data.

3.Performance Variability: The performance of DeNN may vary significantly depending on the dataset and the specific characteristics of the temporal information being processed. For instance, while it performs well on certain datasets, such as CIFAR10, it does not achieve the same level of accuracy compared with other models, indicating a potential inconsistency in its effectiveness across different tasks.

**Questions:**

1.Challenges with Computational Costs: Despite its strengths, DeNN faces challenges, particularly regarding the computational cost of exponential functions on hardware devices. This limitation can restrict its practical applications in environments where computational resources are limited. Additionally, the memory costs associated with long sequences can be a drawback, especially for applications requiring extensive temporal data processing. The efficiency of active parameters should be indicated by more reasonable computation and description.

2.Complexity in Learning Dynamics: The introduction of synaptic delays adds complexity to the learning process, which may complicate the backpropagation algorithm. This complexity can hinder the model's mathematical analysis and may pose challenges in training. Please explain with more details.

---

### Official Review · Reviewer_aa5H · 2024-11-04

**Soundness:** 3
**Presentation:** 2
**Contribution:** 3
**Rating:** 5
**Confidence:** 3

**Summary:**

The authors introduce a new feedforward spiking delay neural network (DeNN) that can be trained with backprop to competitive task accuracy on MNIST, CIFAR-10, N-MNIST, DVS-Gesture, and GSC. Critically, DeNN does not have weights; instead DeNN only has synaptic delays. In contrast, existing SNN works generally have weights and synaptic delays.

The spiking neurons of DeNN only spike once, and this spike time is a function of the arrival times of presynaptic spikes. The authors choose the spike time function to be a summation of presynaptic spike times weighted by a temporal kernel (later spikes contribute less than earlier spikes). The authors represent each synaptic delay via a Gaussian function whose argument is a learned parameter. The Gaussian function always produces positive input, as delays are positive, and the sign of the learned parameter indicates that postsynaptic connectivity is inhibitory (negative) or excitatory (positive). The authors then introduce a standardization procedure, which could possible be used to have the spike time function ignore late spikes.

The authors introduce an input processing methodology for converting event-based datasets into coarser-grained frames with equal numbers of events in each frame. Importantly, this preprocessing step generates frames such that the timing information of the events within each frame is compressed in some fashion so that some timing information is still available to the DeNN.

The authors define a convolutional architecture for the tasks listed above and add a learnable long-term timing memory input preprocessing stage for the GSC task. The authors achieve competitive performance across all tasks and with relatively modest compute/memory cost compared to other baseline DNNs.

**Strengths:**

Significance.
Delays are potentially a powerful computational primitive in the brain. This work explores the expressive power of delays by training a neural network with only delays. Such exploration highlights what delays are capable of, and this knowledge could be valuable to our understanding of the brain and for building efficient neural networks on novel hardware which supports delays.

Originality.
This is the first work I know of that trains an SNN with backprop with ONLY delays (no weights).

Quality.
This work tests DeNNs on many benchmark tasks and against many baselines.

Clarity.
The equations in the text are helpful to understanding the work.

**Weaknesses:**

In this section, I describe the two key weaknesses that cause my rating to be “marginally below acceptance threshold.” Given substantial improvement in the following two weakness areas, I would consider improving my rating; however, I am unsure whether the necessary changes would be too substantial to be completed in the short rebuttal period.

((1)) The key weakness of this paper is lack of clear exposition. Here are several specific points that illustrate some opportunities to improve the clarity of the exposition.

Line 107, what is an “analog” neuron? Analog hardware? Real-valued?

Line 144, first the authors say that they use delays instead of weights, and then they compare large weights to large delays. The reader is left thinking “I thought we were not using weights, so what am I comparing large delays to here, if weights don’t exist?”

Line 154, do the neurons in DeNN only have a single spike time? (Only spike once?) Is this implied here by saying “the” spike time? Also, is time discrete or continuous here? Would be beneficial to clearly state DeNN is a single-spike network.

Line 170, how does the -\kappa (t_i+1) term represent an incompressible delay?

Line 170, where is the discontinuity at d_ij = 1?

Line 182, I know what a ReLU is, but what is a ReLU-like function? Or stated differently, how is what you are describing in this paragraph achieving a ReLU-like function?

Line 183, what is “some value”?

Line 185, what’s the significance of “almost equivalent”? Does it matter that it’s not exactly equivalent?

Line 185, is q dynamically computed or specified before inference?

Eq 5, what is the definition of the “std” function? I inferred it is likely subtracting mean and dividing by standard deviation, but I do not know because std is not clearly defined.

Line 231/Figure 2, I struggle to understand several aspects of the preprocessing stage, given the current exposition. For instance, why 2rN? Do I have two images, one for positive and negative cells? Do events arrive in continuous time, discretized time? In Figure 2, is “N” the same “N” as in “N total pixels in the image”? What is \delta t? Are all \delta ts the same? What is “T”?

Figure 1, do neurons in the next layer have to wait until all neurons in the previous layer have spiked (in the q=1 case)? This seems like a major limitation of the DeNN methodology that should be clearly stated. A further improvement could be addressing the following - if all time indeed needs to be processed in a previous layer before the next layer can start its processing, how is a delay-only network substantively different than a single time-step weight/activation network? Spoken alternatively, it would be very valuable to understand the connection between conventional weight/activation DNNs if I consider delays as weights and spike times as activations. E.g., where would the aforementioned ReLU-like nature come in?

Line 301, rN is not 2rN?

Eq 10, it would be helpful to give more context/verbal exposition regarding the high-level functioning of Eq 10.

Figure 5, “Bottom line…” I understand that this figure shows differences between frames, but what is the purpose of showing this in the context of the paper’s narrative? When I was reading, I did not get the “bottom line” when the “bottom line” was highlighted, and I was confused as to what the bottom line is.

Table 1, what are the units of computation? FLOPS? Such a unit would be more informative than “# comp.”

Figure 3, what is the correct class in this case?

Line 421, what does it mean for the activity of neurons to accelerate? I understand, e.g., that objects accelerate in physics, but I don’t understand the correspondence with neural activity acceleration. Is there, for instance, a second derivative in the neural activity?

Line 470, “train cars”; do the authors mean training for automated driving?

Figure 6, I would recommend labels for left/right panels in the figure caption.


((2)) The secondary weakness of this paper technical incorrectness. Specifically, the authors make several statements that are too strong and/or not true. Highlighting a few examples to illustrate this general weakness:

For example, Line 30, “these networks [DNNs] are [not] energy efficient” – I can understand why one would want to improve the efficiency of DNNs or aspire to reach the energy efficiency of the brain, but to say DNNs are not efficient as a whole class ignores a lot of work on efficient DNNs out there. I’d suggest a less strong statement than “these networks are not energy efficient”; something like “DNNs can use a lot of energy”.

Another example is Line 50, “temporal plasticity is essential to treat temporal information” is not true. For example, one can treat temporal information with RNNs that do not have temporal plasticity.

Another example is Eq 8: later in the paper, Adam is used as an optimizer, and Eq 8 is not the equation for Adam.

Another example is Line 207, “event-based datasets consist of images from event-based cameras…” Not all event-based datasets are image datasets. And Line 2010, “the number of images and events is huge” – the total number of events does not change. The number of images may get huge, but the number of events does not get huge.

**Questions:**

The questions that follow are more minor compared to the weaknesses communicated in the above “Weaknesses” section.

How do delays compare to synaptic time constants? For example, in LIF neurons models, often post-synaptic input ramps up over time with some time constant and then ramps down. If the ramp-up time is large, this acts similarly to a large delay. Could synaptic time constants act as substitutes to delays?

What are the advantages and disadvantages of SSNs in which neurons spike only once, versus SNNs in which neurons spike multiple times?
Is there anything that can be said about the correspondence between weight pruning (e.g., by magnitude) and your q-value (which can be used to ignore synaptic connections with long delays)?

Overall, thank you for this fascinating work. I am enthusiastic about your exploration here in delay networks and the fascinating technical details, such as the Gaussian on learnable delays to make the delays real, and your input preprocessing methodology. I find your contribution valuable, and I hope you publish an improved version of this work regardless of the ICLR review process.

---

### Meta-Review · Area_Chair_BDao · 2024-12-07

**Metareview:**

This work presents a "Delayed Neural Network" that adapts spiking neural networks to compute based on timing delays rather than absolute values at the inputs.  The method was then applied to a number of datasets. While the reviewers all thought the conceptual framework was interesting and appeared novel, there were a number of concerns raised. Many of the concerns pertained to either 1) the clarity of the manuscript, 2) the precision of the claims, and 3) the variability and consistency of the results. The authors responded to these concerns at length, however in reading the replies, I feel that these are significant changes that will require more than the allotted rebuttal time to properly integrate. I feel that a more careful amending of the manuscript would benefit a future submission.

**Additional Comments On Reviewer Discussion:**

Unfortunately the reviewers did not respond despite the effort by the authors to respond. There was also a concern by the authors on one review that was a bit more cursory than the others, and I discounted that review in my assessment, focusing on the more in-depth reviews.

---

### Decision · Program_Chairs · 2025-01-22

Reject